# Two doses of SARS-CoV-2 vaccination induce robust immune responses to emerging SARS-CoV-2 variants of concern

Donal T. Skelly [1,2,3,34], Adam C. Harding[4,34], Javier Gilbert-Jaramillo [4], Michael L. Knight [4], Stephanie Longet [5,6], Anthony Brown [1], Sandra Adele[1], Emily Adland[7], Helen Brown[1], Medawar Laboratory Team*, Tom Tipton [5,6], Lizzie Stafford[8], Alexander J. Mentzer [6,8], Síle A. Johnson[3,9], Ali Amini [1,3,10], OPTIC (Oxford Protective T cell Immunology for COVID-19) Clinical Group*, Tiong Kit Tan [11], Lisa Schimanski[11,12], Kuan-Ying A. Huang[13], Pramila Rijal[11,12], PITCH (Protective Immunity T cells in Health Care Worker) Study Group*, C-MORE/PHOSP-C Group*, John Frater [1,3], Philip Goulder[7], Christopher P. Conlon [8], Katie Jeffery [3], Christina Dold[14,15], Andrew J. Pollard [14,15], Alex Sigal[16,17,18], Tulio de Oliveira[17,19,20,21], Alain R. Townsend [11,12], Paul Klenerman[1,3,10,15], Susanna J. Dunachie [1,3,22,23], Eleanor Barnes [1,3,10,15,35], Miles W. Carroll[5,6,35] & William S. James [4,35✉]

The extent to which immune responses to natural infection with severe acute respiratory syndrome coronavirus 2 (SARS-CoV-2) and immunization with vaccines protect against variants of concern (VOC) is of increasing importance. Accordingly, here we analyse antibodies and T cells of a recently vaccinated, UK cohort, alongside those recovering from natural infection in early 2020. We show that neutralization of the VOC compared to a reference isolate of the original circulating lineage, B, is reduced: more profoundly against B.1.351 than for B.1.1.7, and in responses to infection or a single dose of vaccine than to a second dose of vaccine. Importantly, high magnitude T cell responses are generated after two vaccine doses, with the majority of the T cell response directed against epitopes that are conserved between the prototype isolate B and the VOC. Vaccination is required to generate high potency immune responses to protect against these and other emergent variants.

A full list of author affiliations appears at the end of the paper.

The emergence of new lineages of SARS-CoV-2 on three continents towards the end of 2020, and their rapid expansion at the expense of the previously dominant lineages, poses significant challenges to public health[1]. In order to address these challenges effectively, there is an urgent need to understand the biological consequences of the mutations found in these lineages, and the consequential impact on their susceptibility to current control measures, particularly vaccines.

In early 2021, three variants B.1.1.7 (Alpha), B.1.351 (Beta) and P.1 (Gamma) were identified as variants of concern (VOC[1]). These three variants share the N501Y substitution in the receptor-binding domain (RBD) of spike glycoprotein (S), which increases the binding affinity of S with the virus's cellular receptor, angiotensin-converting enzyme 2 (ACE2)[2] (see Fig. 1). As of 1 March 2021, N501Y is present globally in 77% of currently sequenced samples[3]. Lineage B.1.1.7, first identified in the UK in September 2020, is characterized by additional mutations in S, such as deletion of residues 69 & 70 and the P681H substitution, for which plausible effects on the virus biology are proposed, as well as five other mutations in S, a premature stop codon in ORF8, three substitutions and a deletion in ORF1 and two amino acid substitutions in nucleoprotein (N), of as-yet unknown significance. Lineage B.1.351[4] was first identified in November 2020 in South Africa and is characterized by two additional substitutions of likely significance in RBD, namely, K417N and E484K. The former is predicted to disrupt a salt bridge with D30 of ACE2, a characteristic of SARS-CoV-2 in distinction to severe acute respiratory syndrome coronavirus (SARS-CoV-1), but may not impact on binding, whereas the latter, which might disrupt the interaction of RBD with K31 of human ACE2, may enhance ACE2 binding[2,5]. On 1 March 2021, this lineage accounted for 5% of all current sequences globally, and 100% of those identified in South Africa. The third variant of concern, P.1 (formerly B.1.1.28.1) is characterized by K417T, in addition to E484K and N501Y, and accounted for 80% of all viruses sequenced in Brazil on 1 March 2021. In early 2021, E484K had been detected first in lineage B.1.1.7 in the United Kingdom (UK)[6] and subsequently in lineages A23.1, B.1 and B.1.177, as well as in imported cases of B.1.51 and P.2[1]. Our data confirm that VOC, particularly those such as B.1.351 with substitutions at residues 484 and 417,

escape neutralization by antibodies directed to the ACE2-binding Class 1 and the adjacent Class 2 epitopes but are susceptible to neutralization by the generally less potent antibodies directed to Class 3 and 4 epitopes on the flanks of the RBD. A further rapidly spreading isolate, was recognised as a VOC in May 2021. B.1.617.2 (Delta) was first isolated in India and also shows some evidence of immune escape, specifically from neutralizing antibodies, but to a lesser degree than B.1.351[7].

The immune correlates of protection against infection and disease caused by SARS-CoV-2 are imperfectly understood (reviewed by[8,9]). Classically, neutralization by antibody, measured by reduction in plaque or infectious foci by authentic virus in vitro is considered a major component of protection. Antibodies may also offer protection via fragment crystallizable (Fc)–Fc receptor interactions[10] and harnessing of innate immune function. Diverse antibody-dependent macrophage, neutrophil, complement and natural killer cell functions have been demonstrated after SARS-CoV-2 infection and vaccination[11–13]. Recent studies have demonstrated that symptomatic re-infection within six months after the first wave in the UK was very rare in the presence of anti-S or anti-N IgG antibodies[14,15]. Virus-specific lymphocytes may play an important direct role in protection, in addition to their indirect role in supporting and driving development of antibody-producing cells. Robust T cell immune responses to S, M, N and some ORF antigens are readily detected after infection (with CD4 positive cells dominating), correlate with disease severity and are durable for at least several months[16–18]. Furthermore, CD8 depletion studies in non-human primate (NHP) challenge studies suggest T cells also play a protective role especially when antibody levels are low[19,20,21]. Nevertheless, passive infusion of neutralizing antibody has been shown to be sufficient to mediate effective protection against SARS-CoV-2 in these NHP studies[20]. Although studies in NHPs of both adenovirus-26 and DNA-based vaccine candidates found that levels of neutralizing antibodies but not of T cells were significantly correlated with viral clearance[19,22], recent reports involving subunit vaccine candidates in NHP found not only neutralizing antibodies, but also N-specific CD4+ responses were a statistically significant correlate of protection[23].

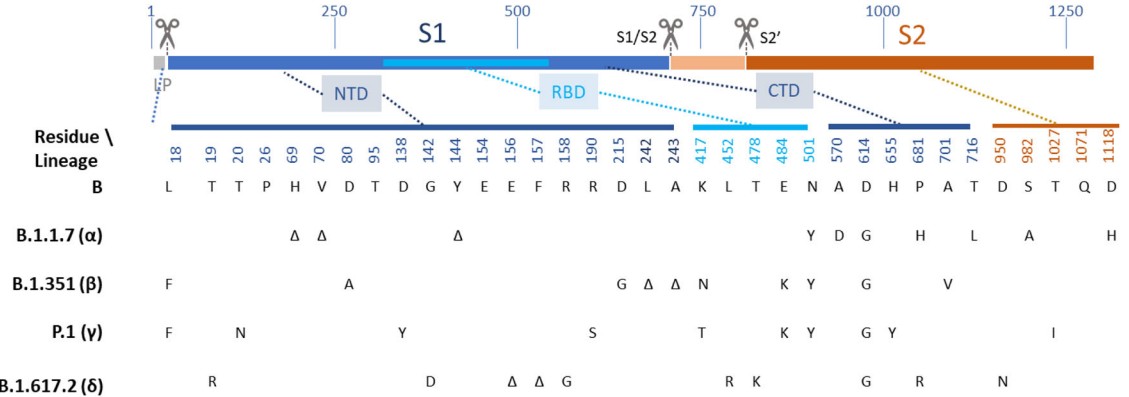

**Fig. 1 Sequence variation in spike glycoprotein.** The open reading frame encoding spike (S) is illustrated, with the position of key features of processing and function indicated to approximate scale (residue number indicated above). During co-translational translocation to the endoplasmic reticulum (ER), the short leader peptide (LP) is proteolytically removed. Following folding, trimer assembly and glycosylation in the ER and Golgi, the *trans*-Golgi localized protease, furin, cleaves the boundary between the S1 and S2 polypeptides. Following binding of the receptor-binding domain (RBD, cyan) to ACE2 on host cells, cell-surface TMPRSS2 proteolytically cleaves the S2' site, facilitating conformational changes to spike that result in fusion of the virus envelope with the plasma membrane. Variant residue positions are indicated below, and their approximate location on the S polypeptide is indicated. Residue identities are shown at each of these positions for a prototype lineage B isolate, and at each position in four lineages of interest, B.1.1.7 (α—Alpha), B.1.351 (β—Beta), P.1 (γ—Gamma) and B.1.617 (δ—Delta), at which the respective lineage differs from prototype. Δ indicates deletion of one or more residues. Note, there are lineage-defining substitutions outside RBD, in the N-terminal domain (NTD) and C-terminal domain (CTD) of S1 (dark blue), and in S2 (tan). These may include changes that directly or indirectly affect antibody-mediated neutralization or cellular immunity, by loss or altered dynamics of epitope, respectively.

Multiple vaccines have been reported to have efficacy against COVID-19 (coronavirus disease 2019) in phase III clinical trials. Of these, three—Pfizer/BNT162b2, Moderna/mRNA-1273 and Sputnik V—that were reported to have efficacies against symptomatic infection in the mid-90% range, had also induced classical neutralizing antibody titres substantially higher than those found on average in convalescent patients[24–26]. In contrast, one—CoronaVac—that showed ~50% efficacy, had been reported to induce neutralizing titres several-fold lower than those found in convalescent patients[27]. The two remaining vaccines, Sinopharm/BBIBP-CorV and AstraZeneca/AZD1222 (ChAdOx-1 nCoV-19), had intermediate values of both clinical efficacy against symptomatic infection and relative potency in generating neutralizing antibody responses[11,28]. mRNA and adenovirus-vectored vaccines generate high magnitude SARS-CoV-2 multispecific CD4+ and CD8+ T cells responses. Reports of vaccines assessed in South Africa where B.1.351 dominates are currently emerging and include Ad26.COV2.S (single dose Ad26 vectored vaccine)[29], Novavax (recombinant spike/adjuvant)[30], AZD1222[31] and BNT162B2[32,33]. With the exception of the recent report on the Pfizer BioNTech vaccine, each of the studies report reduced efficacy in South African populations. Vaccine correlates of protection, and the relative contribution of T cell and humoral immunity, are yet to be precisely defined since detailed immune analysis in people with vaccine breakthrough infections is lacking.

In pseudotype virus neutralization assays, it appears that convalescent sera from patients exposed to prototype strain of SARS-CoV-2, in distinction to vaccine-elicited responses, may not be effective in neutralizing lineage B.1.351[34,35]. As the lineage-defining substitutions include changes in previously identified antibody epitopes and regions of S associated with its processing and rearrangement during cellular infection, this is a very plausible observation.

Here, in order to test whether convalescent sera and sera from vaccine recipients are similarly affected in their ability to neutralize authentic virions, we have undertaken classical neutralization assays against reference isolates of both B.1.1.7 and B.1.351 compared to the early pandemic B isolate. We find that, while cross-neutralization of B.1.1.7 is only modestly reduced compared to that of the prototype B lineage, cross-neutralization of B.1.351 may be markedly reduced in convalescent sera, and after a single vaccine dose. However, both the neutralization of VOC and the generation of virus-specific T cells, is significantly enhanced by a boost vaccination. In addition, vaccination not only induces enhanced reactivity to S from endemic human betacoronaviruses, but also results in significant cross-reactivity to both SARS-CoV-1 and Middle East respiratory syndrome-related coronavirus (MERS-CoV).

Since viral mutations may also affect T cell recognition, we also evaluate the contribution of T cells that target epitopes located at sites of amino acid substitution in the spike glycoproteins of VOC. We show that the majority of T cell responses in recipients of two doses of the BNT162b2 vaccine are generated by epitopes that are invariant between the prototype B lineage virus and VOC. The T cell data are encouraging, and although the weakening of neutralizing antibody titre against VOC might suggest that further additional SARS-CoV-2 vaccinations with reformulated antigen might be required in the future to address new variant lineages[36], our data support the contention that boosting with the current vaccines may well provide sufficient protection.

## Results

### Spike protein sequence differences in SARS-CoV-2 lineages.
The primary structure of the spike glycoprotein (S), and the characteristic sequence variants of the current three lineages of concern are illustrated in Fig. 1. In this study, we analysed the homotypic neutralization of the prototypic, PANGO lineage B isolate, VIC001 (hereafter referred to simply as 'B'), by mAbs, sera from convalescent individuals following SARS-CoV-2 infection, and recipients of the BNT162b2 (Pfizer) vaccine, which are each induced by prototypic S antigen. We then assessed heterotypic neutralization of two VOC (B.1.1.7 and B.1.351). In Fig. 1, we indicate the residues of S at which the respective lineage—as well as two further lineages of concern, P.1 and B.1.617.2—differ from lineage B.

### SARS-CoV-2 binding antibodies and ACE2-spike binding inhibition.
We probed the antibody-binding properties of sera from vaccinated, convalescent and pre-pandemic control sera using a customised Mesoscale Discovery (MSD) coronavirus antigen immunoassay (Fig. 2). We observed that sera from individuals receiving two doses of the Pfizer vaccine showed a non-significant increase in binding to SARS-CoV-2 spike and RBD compared to those receiving single dose and a significant difference from sera of convalescent individuals one month after infection (Fig. 2a and b, respectively, $P < 0.0001$ in all cases by Kruskal–Wallis one-way ANOVA with Dunn's multiple comparisons tests). The absence of N binding in vaccinees (Fig. 2c) supports the designation of these individuals as SARS-CoV-2 unexposed, although it does not prove absence of the previous infection.

There was significant antibody binding to both SARS-CoV-1 and MERS-CoV spike protein in vaccinated and COVID-19 convalescent individuals compared to the pre-pandemic control sera (Fig. 2d and e, respectively). This was particularly marked for SARS-CoV-1 reactivity in fully vaccinated individuals, suggesting that the vaccine can induce a broad response to widely shared epitopes, such as those exemplified by EY 6A[40] and CR3022[45].

We also screened for antibody binding to the spike antigen of the four common circulating coronaviruses (Fig. 2f–i). There is a significant increase in binding to the Betacoronavirus clade A isolates, HCoV-HKU1 and HcoV-OC43, in vaccinated and COVID-19 convalescent sera ($P < 0.0001$) compared to unvaccinated naive sera. Binding to the Alphacoronavirus isolates, HcoV-229E and, to a lesser extent, HcoV-NL63S was also greater in the vaccinees, but not in convalescent sera.

As a surrogate to neutralization, we assessed the ability of sera to inhibit ACE2-spike binding using MSD plates printed with spike proteins representing the prior circulating B lineage, and the more recently evolved VOC (B.1, B.1.1.7, B.1.351 and P.1). Figure 2j indicates that serum from vaccinated individuals receiving either single or double vaccination was able to inhibit ACE2 binding of SARS-CoV-2 spike. The inhibitory effect was significantly higher, ($P < 0.001$ by Mann–Whitney comparisons) in those sera derived from individuals sampled after receiving the boost vaccination compared to post-prime samples. Fold changes in mean inhibitory activity between post-prime and post-boost ranged from 49 for B.1 to 18 for B.1.351. Following vaccine boost, the mean inhibitory activity of B differs significantly from B.1.351 and P.1 but not B.1.1.7 (Friedman test, $P < 0.0001$).

### Neutralization by monoclonal antibodies and reference plasma.
We made use of a panel of six, epitope-mapped neutralizing monoclonal antibodies (NmAbs, Fig. 3a,)[42,43,46,47] in order to map the neutralization sensitivity of VOC to changes in RBD epitopes. We have devised a 'squirrel' diagram to help visualise the binding sites of the various mAbs on the RBD (Fig. 3a). One NmAb, FI 3A, a Class 1 RBD monoclonal antibody (binds to the left side of the head of the squirrel), whose homotypic half-maximal inhibitory concentration (IC50) is of the order of 1 nM, is largely unaffected by the changes in B.1.1.7

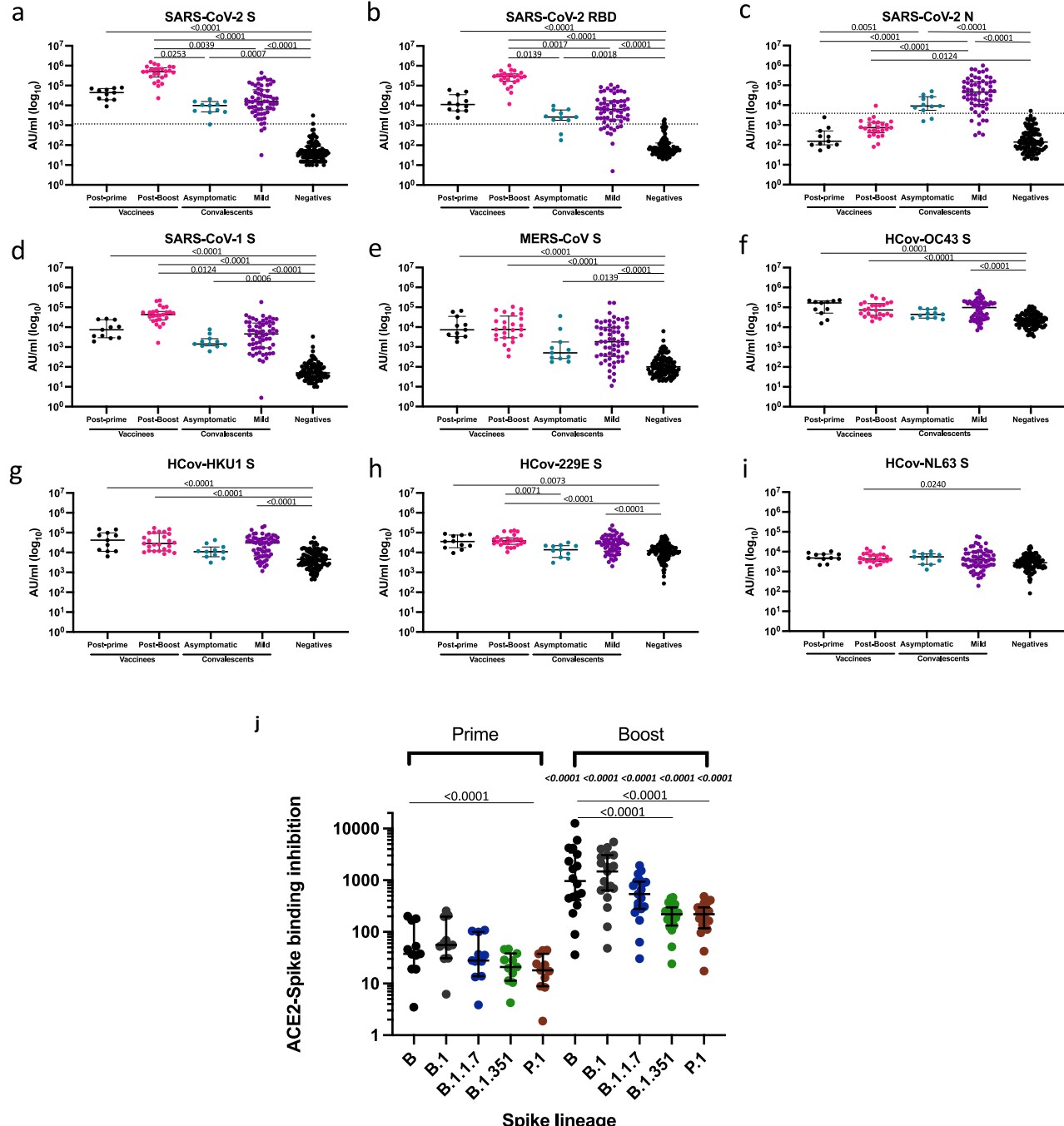

**Fig. 2 Binding assays.** IgG antibodies specific to; **a–c** SARS-CoV-2 (spike [S], receptor-binding domain [RBD], nucleocapsid [N]), **d**, **e** SARS-CoV-1 S, MERS-CoV S, **f–i** HCoV-OC43 S, HCoV-HKU1 S, HCoV-229E S, HCoV-NL63 S, were measured using an MSD technology platform customised array. Sera analysed were from vaccinees (post-prime and post-boost), asymptomatic (mean 27 days post-PCR positive test, range 22–33 days) and mild COVID-19 convalescent sera (mean 29 days post-symptom onset, range 18–40 days) and a cohort of prepandemic sera collected between 2014 and 2018 negative for SARS-CoV-2 (negatives). Data are displayed as calculated concentrations which use an MSD standard reference curve to interpret arbitrary units (AU). Statistical difference between the groups was performed using a two-tailed Kruskal–Wallis one-way ANOVA with Dunn's post-test for multiple comparisons made to compare all groups (significant adjusted $P$ values are displayed). Vaccinees post-prime $n = 11$; vaccinees post-boost $n = 25$; negatives $n = 103$; asymptomatic COVID-19 convalescents $n = 11$; mild COVID-19 convalescents $n = 62$. The dashed lines in **a–c** show the cut-offs determined as the mean of negatives +3 SD. Samples were run in monoplicate (convalescent samples) or duplicates (vaccine samples). **j** Inhibition analysis between ACE2 and recombinant spike from the designated homotypic and heterotypic lineages. Sera derived from individuals receiving prime or boost vaccination: post-prime $n = 11$, post-boost $n = 18$–25 dependent on spike variant. Differences between B and other variants in post-prime and post samples were tested using a Friedman statistical test with Dunn's multiple comparison test (significant $P$ values are displayed on lines linking B to variant). A two-tailed Mann–Whitney $U$ test was used to compare post-prime and post boost groups for each variant. $P$ values are displayed on the boost panel, immediately above each variant and are italicized. Plots show median with error bars indicating ± intraquartile range) (IQR). ACE2 inhibition assay samples were run in monoplicate.

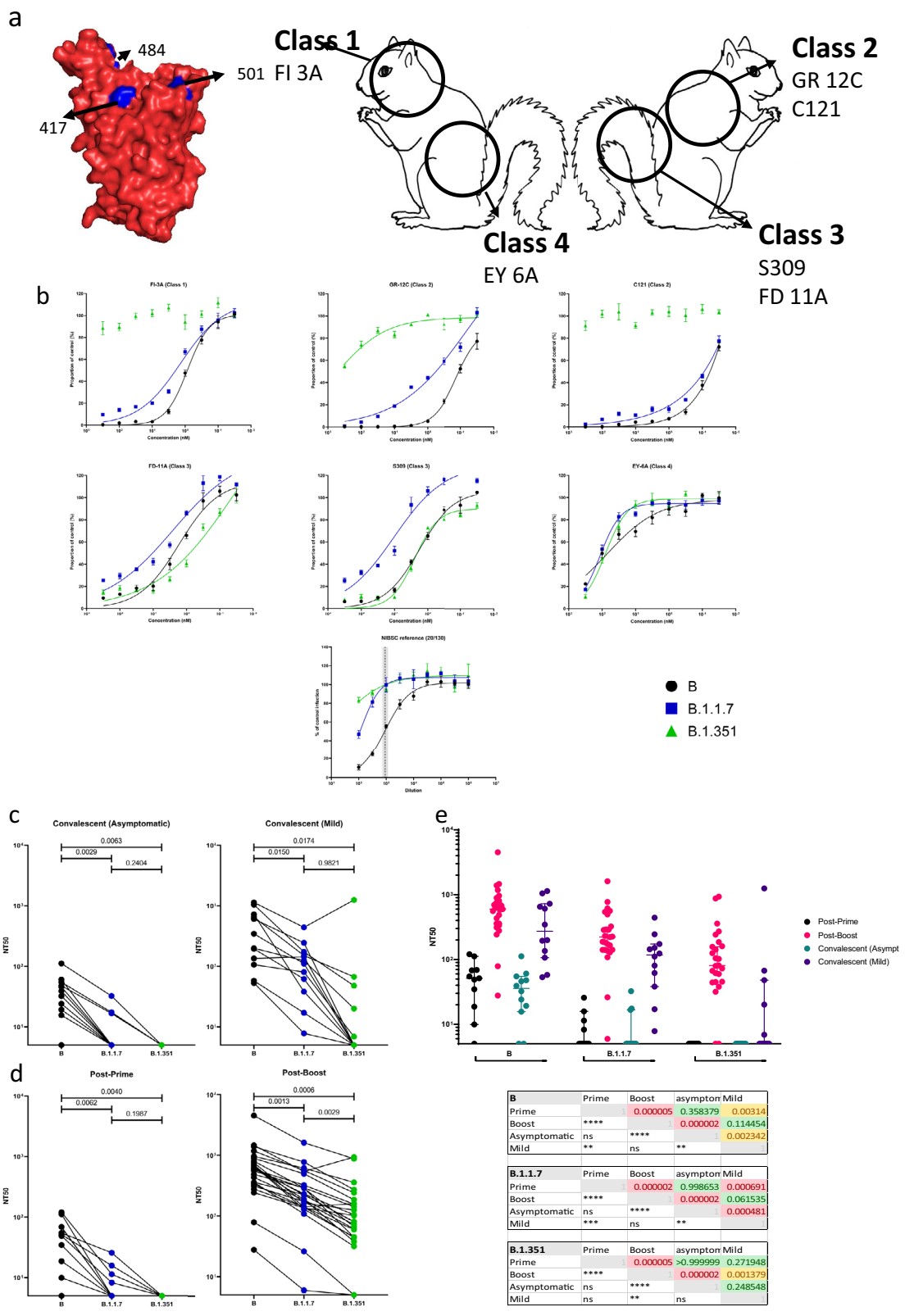

(IC50 = 1.365 nM) but does not neutralize B.1.351. Two other NmAbs, GR 12C and C121, that are Class 2 RBD binding mAbs (binding to the right side of the head of the squirrel), and which have homotypic IC50 ~0.1 nM, show some reduced effectiveness in neutralizing B.1.1.7 and have lost almost all potency against B.1.351. This might be expected, as class 2 antibodies bind to an epitope that includes residue 484 (reviewed by refs. [47],[48]). In contrast, NmAb FD 11A and S309, which are Class 3 RBD mAbs, that bind to the right haunch of the squirrel, and EY 6A, Class 4 monoclonal antibody, that binds to the left haunch of the squirrel, appear to be unaffected by the mutations in the VOC. Polyclonal responses generated by different individuals to natural infection or in response to vaccination may include a varying proportion of antibodies to these and other neutralization epitopes.

**Fig. 3 Homotypic and heterotypic neutralization of key SARS-CoV-2 lineages by antibody.** The potency of neutralization was determined by a focus-forming unit microneutralization assay against authentic virus of prototype B lineage and isolates of B.1.1.7 and B.1.351. **a** A cartoon of a 'squirrel' illustrating the receptor-binding domain (RBD). Markings on the squirrel show the epitopes of the RBD antibodies used in this study (classes 1–4[47]). Neutralization by the panel of monoclonal antibodies binding to four distinct epitopes of RBD (upper left). A space-filling model of prototype RBD (PDB 6YZ5) created in PyMOL, shows the residue of the mutations present in the B.1.1.7 and B.1.351 lineages in blue (upper middle and upper right; same aspect and reverse aspect as the space-filling model, respectively). **b** Neutralization by the international reference plasma NIBSC 20/130[38], nominally NT50 = 1/1000. The mean number of foci ± SD relative to a no-antibody control is plotted against the reciprocal of the respective serum dilution. Data were fitted by non-linear regression in GraphPad Prism 9 to the Hill Equation, with TOP and BOTTOM constrained to 100 and 0%, respectively. Where a significant fit was obtained, it is represented by a trend line on the respective plot, and NT50 values used in Supplementary Fig. 1 and main results. NT50 against B established in our assay indicated by the vertical dashed line with grey bars indicating the 95% confidence interval. **c** Neutralization by convalescent sera from asymptomatic participants (left) and those with mild symptoms (right) against B, B.1.1.7 and B.1.351 isolates. A Tukey's multiple comparison test was performed for each serum group, comparing mean NT50 response to each virus isolate. P values of all comparisons are displayed in the figures. **d** Neutralization by sera from recipients of a single dose (left) and both prime and boost doses (right) of BNT162b2 vaccine. **e** Homotypic and heterotypic neutralization potencies of the three sources of antibody against the three isolates, shown by individual (dots) and sub-population mean and SD of NT50 values shown with error bars (upper panel). For each isolate, pairwise comparisons of average NT50 estimates were made between groups of serum using the Kolmogorov–Smirnov non-parametric test. P values for the r statistic are shown (lower panel), both numerically and symbolically. P > 0.05 in green and 'ns'. No results for 0.05 > P > 0.01. 0.01 > P > 0.001 in yellow and **P < 0.001 in red and ****. Vaccinees post-prime n = 11; vaccinees post-boost n = 25; asymptomatic COVID-19 convalescents n = 12; mild COVID-19 convalescents n = 13. MNA tests were performed in quadruplicate for all samples. Each plate contained serum-free controls for normalization of results.

Figure 3b shows the performance of NIBSC 20/130 plasma on our MNA. The use of this international standard enables calibration of assay sensitivity with other published works and serves as a reliable positive control. The expected NT50 of this standard against the B isolate, as stated in the accompanying data sheet, is 1:1280. Our own result demonstrates consistency with this expectation (918.2; 95% CL 729.6–1165). Of note, neutralizing activity of this plasma is substantially diminished against both B.1.1.7 (125; 95% CL 86–164), and B.1.351 (14; 95% CL 0.1–51). Serum derived from non-infected, unvaccinated individuals acted as negative controls. These naïve samples demonstrate the absence of any background neutralization (Supplementary Fig. 2a).

**Neutralization by sera from COVID-19 convalescents.** Sera from convalescent individuals neutralized prototype B virus with highly variable potency (NT50 range <5 to 1140, Fig. 3c and e), though sera from those with mild symptoms were significantly more potent on average than those with asymptomatic infection (NT50 438.4 and 38.5, respectively, P = 0.002). Neutralization titres against B.1.1.7 were below the limit of detection in 9/12 asymptomatic convalescent individuals but were detectable in all those with mild symptoms. The neutralizing potency of mild convalescent sera against B.1.1.7 was significantly greater than that of asymptomatic sera (NT50 133 and 9.3, respectively; Kolmogorov–Smirnov test, P = 0.0005).

The decline in neutralization potency was more marked against the B.1.351 isolate, with convalescent sera from 12/12 asymptomatic and 7/12 mild having undetectably low neutralizing potency. Although there was no significant difference between the mean NT50 of mild versus asymptomatic sera against B.1.351 (119 and <5 respectively, P = 0.25), the reduction in potency overall in relation to prototype B virus was very significant (P = 0.000003).

**Neutralization by sera from vaccine recipients.** After a single dose of BNT162b2 vaccine, homotypic neutralization potency was on average comparable to that of an asymptomatically infected cohort (NT50 53.8 and 38.5, respectively, P = 0.36), but lower than sera from those who had recovered from mild infection (NT50 438.3, P = 0.003; see Fig. 3d and e). Neutralization after one dose was undetectable against B.1.1.7 in 7/11 samples, and in all 11 sera tested against B.1.351.

Sera drawn between 7 and 17 days after a second dose of BNT162b2 vaccine—administered 18–28 days after the first dose—

neutralized lineage B virus with high potency (average NT50 = 768) and 23/25 individuals had NT50 ≫ 1/100, (Fig. 3d), whereas 2/25 individuals showed more modest titres (10 < NT50 < 100). These sera neutralized the B.1.1.7 isolate with a significantly lower potency (average NT50 = 320; P < 0.0001, Kolmogorov–Smirnov test); the same 23/25 had NT50 titres > 100 and 2/25 NT50 titres 10–100. The decline in neutralization potency against the B.1.351 isolate was further significantly reduced (NT50 = 171; P = 0.000001), but 12/25 retained NT50 titres > 100, 11/25 NT50 10–100 with only the 2/25 with modest homotypic neutralization potency having undetectable heterotypic neutralizing potency.

The relationship of the neutralizing titre of each individual's serum to B to the corresponding titre against each variant apparent in Fig. 3d is significant. Spearman correlation coefficients (r) are: 0.76 (B to B.1.1.7, CL 0.52–0.98; P = 0.0000092); 0.74 (B to B.1.351, CL 0.48–0.88. P = 0.00002); and 0.79 (B.1.1.7 to B.1.351, CL 0.57–0.91, P = 0.000002).

**T cell responses to spike antigens in B strain and VOC.** Following two doses of BNT162b2, spike-specific T cells were detected in all individuals against spike antigens covering the prototypic B strain, assessed in IFN-γ ELISpot assays peaking 7 days after the second vaccine (mean magnitude 561, range 110-1717 SFC/10^6 PBMC) (Fig. 4a and Supplementary Fig. 4). Spike-specific T cells could not be detected in unvaccinated SARS-CoV-2 unexposed HCW (Supplementary Fig. 3). This is in keeping with our previously published work[14] demonstrating our highly specific IFN-γ ELISpot assay, which yields negligible T cell responses detected to specific peptides in unexposed subjects with the selected peptide concentrations and incubation time.

Assessing the contribution of T cells that target epitopes located at the site of B.1.1.7, B.1.351 and P.1 spike mutation sites, we find that T cells target epitopes spanning mutation sites in 18/24 individuals (Fig. 4b). In each individual, T cells targeted 0–19 (mean 6) epitopes located at mutation sites (Supplementary Table 2) with a total of 8, 9 and 10 epitopes targeted in lineage B.1.1.7, B.1.351 and P.1 respectively. The overall contribution of T cells targeting mutation regions to the total spike specific response is (mean and range) 13% (0–67%) for B.1.1.7, 14% (0–44%) for B.1. 351 and 10% (0–29%) for P.1 (Fig. 4c). Although the overall contribution of T cell responses to mutational regions/total spike responses was low, in general multiple individuals had T cells that targeted each of the mutational regions, spanning all spike domains (Fig. 4d and

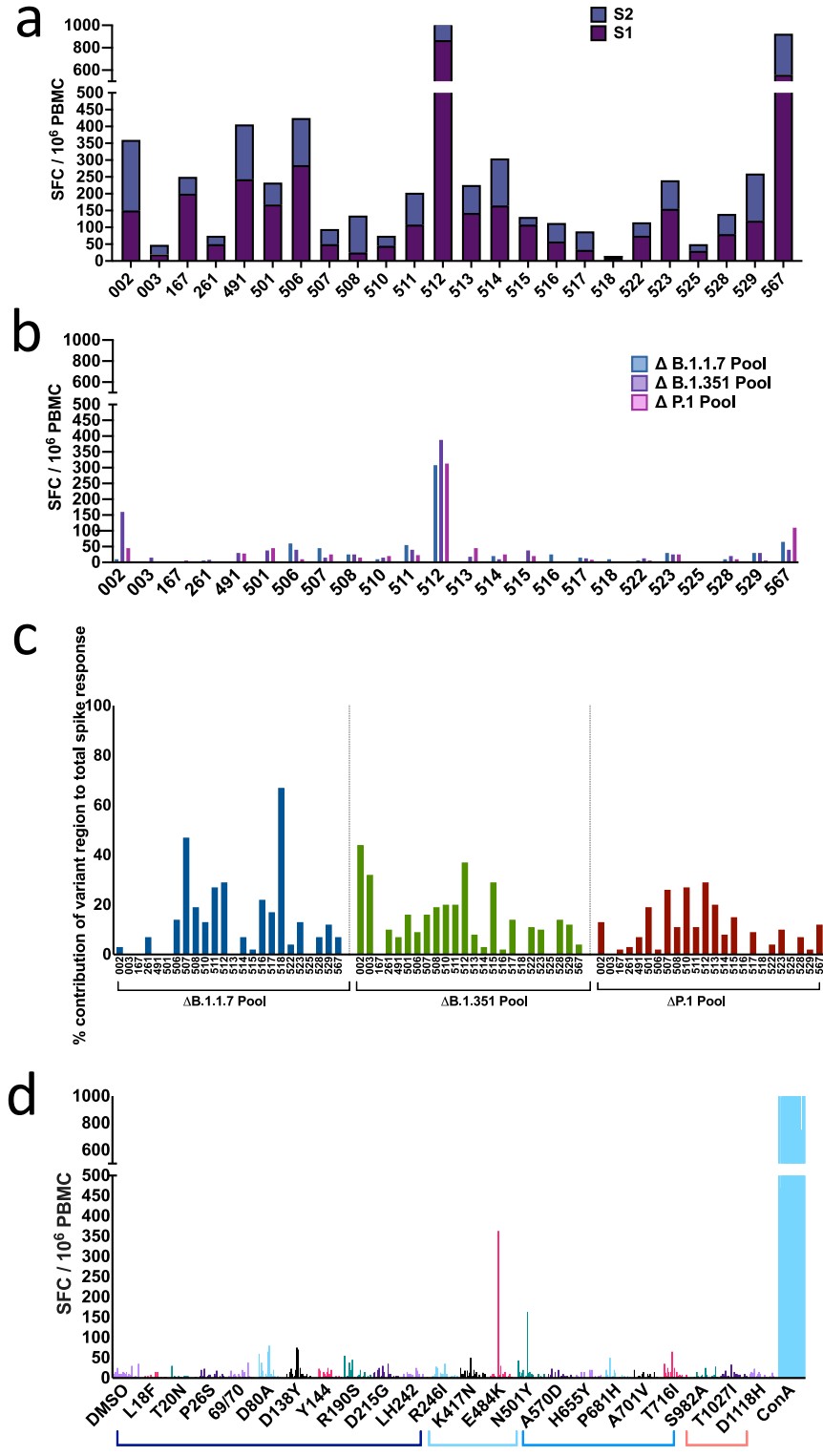

**Fig. 4 ELISpot responses to prototype, B.1.1.7, B.1.351 and P.1.** T cell responses were measured using IFN-γ ELISpot assays in 24 healthy volunteers, 7–17 days after receiving the 2nd dose of BNT162b2. **a** T cell responses to 15–18-mer peptides in B strain overlapping by 10 amino-acids and spanning the entire spike region. **b** Summed T cell responses to peptides from B strain that mapped to sites with mutations in B.1.1.7 ($n = 17$ peptides), B.1.351 ($n = 21$ peptides) and P.1 ($n = 22$ peptides). **c** Percentage contribution T cells (using B peptides) that target mutational regions within B.1.1.7, B.1.351 and P.1, relative to the total T cell spike response in each of the 24 volunteers. **d** T cell responses to 22 individual peptides in B strain that have corresponding mutations in B.1.1.7, B.1.351 and P.1 variants. Standardised ELISpot assays were run in triplicate for background and spike peptides and duplicates for all others to allow cell preservation. DMSO control with matching percent DMSO was also used in all assays to account for DMSO content in peptide pools. Each bar represents one volunteer with a positive response (defined as a response to the peptide minus the background that was greater than twice the background). SFC/10⁶ PBMC = spot forming cells per million peripheral blood mononuclear cells, with background subtracted.

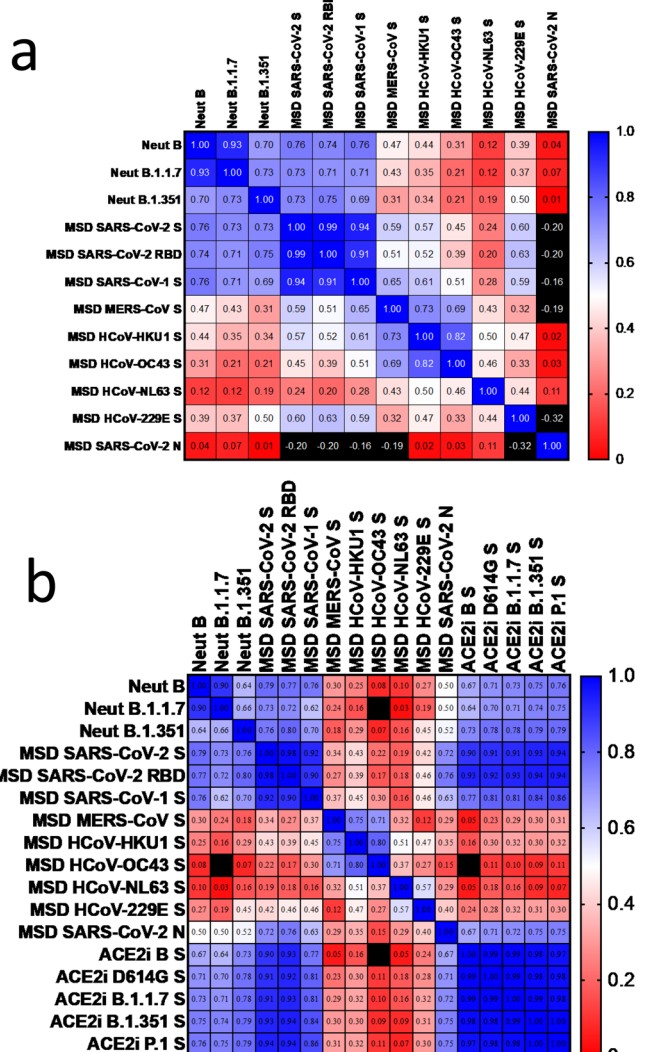

**Fig. 5 Cross-correlation of immune parameters.** For each serum, pairwise Spearman correlation analyses were undertaken between the value of binding of serum antibody to coronavirus antigens, the ACE2-spike binding-inhibition potency (see Fig. 2), and the homotypic and heterotypic neutralizing titre of the same sera (see Fig. 3). **a** Heatmap of two-tailed Spearman's r parameter for each comparison in which spike binding data was available across samples drawn from participants after 1 dose (n = 11, mean 27 days after the 1st dose) and 2 doses of BNT162b2 vaccine (n = 25, mean 8 days after receiving the 2nd dose), and previous infection (n = 20, 11:9 asymptomatic to symptomatic, mean 28 days since PCR test for asymptomatics or symptoms onset). Colour mapping is dual gradient from Blue (r = 1.0) through White (r = 0.5) to Red (r = 0). Values outside this range are Black. **b** Heatmap Spearman's r parameter for each comparison in which ACE2-spike binding-inhibition data were available (n = 35:11 after the 1st dose and 24 after the 2nd vaccine dose). Colour mapping as in (**a**).

Supplementary Table 2). T cell responses to total spike and mutation sites were further assessed in a small number of vaccinees after only a single vaccine; here low magnitude T cell responses were detected (Supplementary Fig. 5a), with T cells targeting mutational regions in 3/5 vaccinees (Supplementary Fig. 5b). Similar to post boost responses, the relative contribution of these to total spike was low (% mean contribution and range; 24% (2–34%) for B.1.1.7, 11% (0–20%) for B.1. 351 and 7% (0–23%) for P.1) (Supplementary Fig. 5c).

**Prediction of neutralization by immunoassay binding.** Authentic virus neutralization assays require specialist staff and facilities that are not widely available, and access to reference isolates of virus that are laborious to distribute. Accordingly, we asked whether high throughput ELISA-style immunoassays could provide a degree of predictive value for heterotypic neutralization following vaccination or previous infection. We performed Spearman non-parametric correlation analysis between the neutralization, spike-binding, and inhibition of ACE2-spike binding results obtained from the same sera, as detailed in the foregoing sections.

The results show that there is a significant correlation (P < 0.0001) between spike-binding and neutralization of authentic virus across samples drawn from participants after 1 and 2 doses of vaccine, and previous infection (Fig. 5a). The Spearman r between neutralization by serum of lineage B virus and the binding activity to lineage B RBD is 0.74 (95% CI 0.6–0.8, n = 56, P = 1 e−10), and the r between neutralization of lineage B.1.351 and binding to B RBD is 0.75 (0.6–0.8, n = 56, P = 3 e−11). These relationships hold for post-boost comparisons (Supplementary Fig. 6), and comparison in convalescents after infection (Supplementary Table 3a but do not hold strongly in post-prime comparisons alone (Supplementary Table 3b). Interestingly, binding activity to SARS-CoV-2 S predicted binding to both SARS-CoV-1 S and MERS-CoV S very well (r = 0.94 (0.90–0.96), n = 56, P = 2 e−27; and r = 0.59 (0.4–0.7), n = 56, P = 1 e−6). Moderate correlations (r of the order of 0.5) were seen with binding to the spike of endemic human betacoronaviruses and HCoV-229E, but not to that of HCoV-NL63.

We assessed inhibition of ACE2-spike binding in the sera of vaccinated participants (Fig. 5b). The correlation between neutralization and inhibition ACE2-spike binding is similar, with r = 0.67 (0.4–0.8, n = 35, P = 0.00007) for lineage B, and r = 0.79 (0.6–0.9, n = 35, P = 2 e−8) for lineage B.1.351 It is important to note that, in this assay, the spike sequences correspond to the virus lineage in the neutralization assay.

No significant correlations were observed between humoral and T cell responses to whole S protein, determined by ELISpot analysis in those who had received two vaccine doses.

## Discussion

Our results show that both binding and neutralization by antibodies induced by the S protein of prototypic lineage B is diminished to S from recent VOC; B.1.351 to a greater extent than B.1.1.7. This broad trend masks both qualitative and quantitative differences in antibody responses by individuals, whose serum may contain differing proportions of antibodies to neutralizing epitopes that we show here are sometimes conserved between lineages.

Given the cost and difficulty of authentic virus neutralization assays, it is encouraging that in our hands, both a high-throughput spike-binding assay and a spike-ACE2 binding-inhibition assay provide a significant correlation with the neutralizing potency—both homotypic and heterotypic—of sera after two doses of vaccine (BNT162b2). It is perhaps not surprising that in post-prime sera, this relationship is much weaker, given our current understanding of the distribution of the neutralization epitopes on the spike glycoprotein (see Fig. 2a) and maturation of the immunoglobulin response. For example, neutralizing antibodies that compete for ACE2 binding tend to lie in class 1 and 2, whose epitopes are subject to VOC-associated substitutions, whereas neutralization epitopes of class 3 and 4 do not overlap with the ACE2 binding site and are conserved in most isolates.

It is also reassuring to find that the majority of T cell responses in recipients of two doses of the BNT162b2 vaccine are generated by epitopes that are invariant between the prototype and two of the current VOC (B.1.1.7 and B.1.351). These data are compatible with a recent report that the sequences of the vast majority of

SARS-CoV-2 T cell epitopes are not affected by the mutations found in the B.1.1.7 or B.1.351 variants[49], with no significant differences observed in CD4 and CD8 responses to a pool of S peptides corresponding to the ancestral sequence and those corresponding to the different variants. T cell responses to SARS-CoV-2 are known to target a wide range of regions in spike[50]. Consistent with this, our data show that neutralization of sera and T cell activity are independent[22]. Moreover, in over 90% of the recipients of two vaccine doses, heterotypic neutralizing titres (NT50) remain comfortably above the level associated with immune protection in non-human primate challenge studies[19,22]. However, in a majority of individuals whose homotypic neutralization titres were more modest—including over 50% of convalescent COVID-19 individuals and recipients of a single dose of vaccine—heterotypic neutralization dropped to negligible levels. This loss of cross-neutralization was particularly notable against B.1.351 with potential implications for vaccine effectiveness in populations where this VOC dominates and when only moderate levels of S antibodies are generated after vaccination.

It should be noted that neutralization escape, observed in a well of a micro-titre plate, is not direct evidence of vaccine failure. Non-neutralizing antigen-specific antibodies, T cells and innate lymphocytes clearly have the potential to contribute to vaccine efficacy[51]. The acceptance that prior infection with influenza virus results in reduced disease against subsequent infection with heterosubtypic strains, in both human and animal challenge studies, provides further evidence that cellular components and non-neutralizing antibodies make an important contribution to protection[52–54]. We also note that the recent South African and UK vaccine clinical trials for Novavax reportedly showed 60 and 85.6% protective efficacy against infection for the B.1.351 and B.1.1.7 VOC, respectively, with no cases of vaccinated individuals requiring hospitalization due to severe disease[55]. Ongoing analysis of real-world vaccine roll out will illuminate the extent of vaccine breakthrough with VOC, although there is already evidence that two-dose regimen of AZD1222 does not protect against mild-to-moderate COVID-19 caused by B.1.351[56].

Nevertheless, our results re-emphasize the urgent need to deploy the most effective vaccine strategies as widely and rapidly as possible in order to provide population protection against the emerging lineages of concern of SARS-CoV-2. Our findings show clearly that the weaker responses generated for example by natural infection or single doses of vaccine, do not provide adequate cross-neutralization. The results support the recommendations by Pfizer, the FDA and EMA for a two-dose vaccine regimen.

## Methods
**Volunteer samples**. Volunteers were recruited at Oxford University Hospitals NHS Foundation Trust in ethically approved studies. Healthcare Workers (HCWs) with asymptomatic SARS-CoV-2 infection, defined as being SARS-CoV-2 polymerase chain reaction (PCR) positive on screening without symptoms (mean 28 days post-PCR testing, range 24–34 days) and mild symptomatic COVID-19, defined as being SARS-CoV-2 PCR positive and having symptoms not requiring O2 support/hospitalization (mean 28 days post-symptom onset, range 24–37 days) were recruited under the OPTIC Study: Oxford Translational Gastrointestinal Unit GI Biobank Study 16/YH/0247 [REC at Yorkshire & The Humber—Sheffield]. HCWs not known to be previously infected with SARS-CoV-2, were recruited after vaccination with the COVID-19 mRNA Vaccine BNT162b2 (Pfizer). 11 participants were recruited post-prime (mean 29 days after a single dose, range 18–41). 25 participants were recruited post-boost (mean 8 days after the second dose, range 7–17 days) and assessed again for T cell reactivity 28 days boost. An additional 13 unvaccinated, non-SARS-CoV-2 exposed HCW were recruited and assessed for T cell reactivity. Four unvaccinated participants were recruited under the Observational Biobanking study approvals SthObs (18/YH/0441) and assessed for neutralizing antibodies. Pre-pandemic negative control sera, used for binding assays, were obtained from a prior vaccine study of the National Vaccine Evaluation Consortium, performed in 2017. Ethics approval from NHS Heath Research Authority—NRES committee London City and East 2017. Supplementary Table 1 shows summary details for each group in terms of days since vaccination or infection and the assays were performed. The study was conducted according to the principles of the Declaration of Helsinki (2008) and the International Conference

on Harmonization (ICH) Good Clinical Practice (GCP) guidelines. Written informed consent was obtained for all patients enrolled in the study.

**Virus isolates**. Prototype isolate (PANGO lineage B) was Victoria/01/2020[37], received at P3 from Public Health England (PHE) Porton Down (after being supplied by the Doherty Centre Melbourne) in April 2020, passaged in VeroE6/TMPRSS2 cells (NIBSC reference 100978), used here at P5, and confirmed identical to GenBank MT007544.1, B hCoV-19_Australia_VIC01_2020_ EPI_ ISL_ 406844_ 2020-01-25. B.1.1.7[38], (20I/501Y.V1.HMPP1) isolate, H204820430, 2/UK/VUI/1/2020, received in Oxford at P1 from PHE Porton Down in December 2020, passaged in VeroE6/TMPRSS2 cells, used here at P4. B.1.351 (20I/501Y.V2.HV001) isolate[39] was received at P3 from the Centre for the AIDS Programme of Research in South Africa (CAPRISA), Durban, in Oxford in January 2021, passaged in VeroE6/TMPRSS2 cells, used here at P4. For all isolates, identity was confirmed by deep sequencing at the Wellcome Trust Centre for Human Genetics, University of Oxford.

**Microneutralization assay (MNA)**. The study was performed in the containment level 3 facility of the University of Oxford, operating under license from the Health and Safety Authority, UK, on the basis of an agreed code of practice, risk assessments (under the Advisory Committee on Dangerous Pathogens guidance) and standard operating procedures. The microneutralization assay determines the concentration of antibody that produces a 50% reduction in infectious focus-forming units (FFU) of authentic SARS-CoV-2 in Vero cells (ATCC CCL-81. Quadruplicate serial dilutions of serum, or monoclonal antibody (20 µL), were preincubated with 100-200 FFU (20 µL) of SARS-CoV-2 for 30 min at room temperature. After pre-incubation, 100 µL of Vero CCL-81 cells ($4.5 \times 10^4$) were added and incubated at 37 °C, 5% carbon dioxide. After 2 h, 100 µL of a 1.5% carboxymethyl cellulose-containing overlay was applied to prevent satellite focus formation. Eighteen (B.1.351) or 23 h (B, B.1.1.7) post-infection, the monolayers were fixed with 4% paraformaldehyde, permeabilized with 2% Triton X-100 and stained for the nucleocapsid (N) antigen or spike (S) antigen, using monoclonal antibodies (mAbs) EY 2A and EY 6A, respectively (both used at 1ug/mL)[40]. After development with a peroxidase-conjugated antibody (1:5000 dilution, cat. no. A0170-1ML, Sigma-Aldrich, Germany) and TrueBlue peroxidase substrate, infectious foci were enumerated by ELISpot reader. Data were analysed using four-parameter logistic regression (Hill equation) in GraphPad Prism 8.3.

**Expression and purification of monoclonal antibodies**. Monoclonal antibodies FI 3A (Class 1), GR 12C (Class 2), FD 11A (Class 3) and EY 6A (Class 4) were isolated from convalescent patients as previously described[40]. In brief, plasmablasts from hospitalised PCR-confirmed SARS-CoV-2 infected patients (day 14 to day 22 post onset of symptoms) were isolated. Freshly separated or thawed PBMCs were stained with fluorescent-labelled antibodies to cell surface markers; Pacific Blue anti-CD3 (clone UCHT1, cat. no. 558117, 420 BD), fluorescein isothiocyanate anti-CD19 (clone HIB19, cat. no. 555412, BD), 421 phycoerythrin-Cy7 anti-CD27 (clone M-T271, cat. no. 560609, BD), 422 allophycocyanin-H7 anti-CD20 (clone L27, cat. no. 641396, BD), phycoerythrin423 Cy5 anti-CD38 (clone HIT2, cat. no. 555461, BD) and phycoerythrin anti-human IgG (clone G18-145, cat. no. 555787, BD). The CD3neg CD19pos CD20neg CD27hi CD38hi IgGpos plasmablasts were gated as single cells.

Sorted single cells were used to produce human IgG mAbs, as previously described[41]. Briefly, the variable region genes from each single cell were amplified in a reverse transcriptase-polymerase chain reaction (RT-PCR: QIAGEN, Germany) using a cocktail of sense primers specific for the leader region and antisense primers to the Cγ constant region for heavy chains and Cκ and Cλ for light chains. The RT-PCR products were amplified in separate PCR for the individual heavy and light chain gene families using nested primers to incorporate unique restriction sites at the ends of the variable gene as previously described[41].

Monoclonal antibodies C121 (Class 2) and S309 (Class 3) were derived from the published sequences[42,43] by gene synthesis (GeneArt). These variable genes were then cloned into expression vectors for the heavy and light chains. Plasmids were transfected into the Expi293F cell line for expression of recombinant full-length human IgG mAbs in serum-free transfection medium. The mAbs were then affinity purified using a MabSelectSure column (Cytiva, USA), according to the manufacturer's protocol and buffer exchanged into 1xPBS using a 10k molecular weight cut off. Amicon Ultracentrifugal Unit.

National Institute for Biological Standards and Control (NIBSC) 20/130 reference plasma was obtained from the NIBSC, UK. It is human plasma from a donor recovered from COVID-19.

**Mesoscale discovery (MSD) binding assays**. IgG responses to SARS-CoV-2, SARS-CoV-1, MERS-CoV and seasonal coronaviruses were measured using a multiplexed MSD immunoassay: The V-PLEX COVID-19 Coronavirus Panel 3 (IgG) Kit (cat. no. K15399U) from Meso Scale Diagnostics, Rockville, MD USA. A MULTI-SPOT® 96-well, 10 spot plate was coated with three SARS CoV-2 antigens (S, RBD, N), SARS-CoV-1 and MERS-CoV spike trimers, as well as spike proteins from seasonal human coronaviruses, HCoV-OC43, HCoV-HKU1, HCoV-229E and HCoV-NL63, and bovine serum albumin. Antigens were spotted at 200−400 µg/mL (MSD® Coronavirus Plate 3). Multiplex MSD assays were performed as per the instructions of the manufacturer. To measure IgG antibodies, 96-well plates were blocked with MSD

Blocker A for 30 min. Following washing with washing buffer, samples diluted 1:500–10,000 in diluent buffer, or MSD standard or undiluted internal MSD controls, were added to the wells. After 2-h incubation and a washing step, detection antibody (MSD SULFO-TAG™ Anti-Human IgG Antibody, 1/200 dilution, cat. no. D21ADF-3) was added. Following washing, MSD GOLD™ Read Buffer B was added and plates were read using a MESO® SECTOR S 600 Reader. The standard curve was established by fitting the signals from the standard using a 4-parameter logistic model. Concentrations of samples were determined from the electrochemiluminescence signals by back-fitting to the standard curve and multiplied by the dilution factor. Statistical analysis was performed using Kruskal–Wallis one-way ANOVA.

A multiplexed MSD immunoassay (MSD, Rockville, MD) was also used to measure the ability of human sera to inhibit ACE2 binding to SARS-CoV-2 spike (B, B.1, B.1.1.7, B.1.351 or P.1). A MULTI-SPOT® 96-well, 10 spot plate was coated with five SARS-CoV-2 spike antigens (B, B.1, B.1.1.7, B.1.351 or P.1). Multiplex MSD Assays were performed as per manufacturer's instructions. To measure ACE2 inhibition, 96-well plates were blocked with MSD Blocker for 30 min. Plates were then washed in MSD washing buffer, and samples were diluted 1:10–1:100 in diluent buffer. Importantly, an ACE2 calibration curve that consists of a monoclonal antibody (cat. no. C01ADG-2) with equivalent activity against spike variants was used to interpolate results as arbitrary units. Furthermore, internal controls and the NIBSC 20/130 international standard were added to each plate. After 1-h incubation recombinant human ACE2-SULFO-TAG™ was added to all wells. After a further 1-h plates were washed and MSD GOLD™ Read Buffer B was added, plates were then immediately read using a MESO® SECTOR S 600 Reader.

**Peptides used in IFN-γ ELISpot assays**. Peptides corresponding to SARS-CoV-2 prototype lineage B isolate, VIC01, 15–18 amino-acids overlapping by 10 amino-acids and spanning the entire spike region, were used in IFN-γ ELISpot assays. Spike peptides were used in two pools (S1 and S2) (Mimotopes, Victoria Australia). Cytomegalovirus, Epstein-Barr virus, influenza and tetanus antigens (CEFT) were used in single pools as positive control antigens (2 µg/mL: Proimmune, Oxford, UK). Single peptides (Mimotopes, Victoria Australia) that mapped to sites containing substitutions in lineages B.1.1.7 ($n = 17$), B.1.351 ($n = 21$) and P.1 ($n = 22$), with reference to B, were used in single peptides or pooled by individual VOC. T cell responses to original B strain peptides covering the areas of known sequence/amino acid mutations/deletions in the VOC (B.1.1.7, B.1.351 and P.1) relative to B are assessed. Three peptides, each of which span a single mutational site/region, were used in these assays: (i) firstly in pools to cover all mutation regions within each VOC and then (ii) mapped to single mutational regions. T cell responses to the peptide pools that span the mutational regions are also assessed alone and in relation to the total T cell response against the entire spike antigen.

**IFN-γ T cell ELISpot assays**. Peripheral blood mononuclear cells (PBMCs) were isolated by density gradient centrifugation using Lymphoprep™ ($p = 1.077$ g/mL, Stem Cell Technologies), washed twice with RPMI (Roswell Park Memorial Institute)-1640 (Sigma, St. Louis, MO, USA) containing 10% heat-inactivated fetal calf serum (Sigma), 1 mM Pen (100 U/ml)/Strep (100 µg/ml) and 2 mM L-glutamine (Sigma) and resuspended in R10 and counted using the Guava® ViaCount™ assay on the Muse Cell Analyzer (Luminex Cooperation). PBMCs were frozen and stored in liquid nitrogen before use.

As previously described[44], 96-well Multiscreen-I plates (Millipore, UK) were coated for 3 h with 10 µg/mL GZ-4 anti-human IFN-γ (Mabtech, AB, Sweden) at room temperature. PBMC were added at $2 \times 10^5$ cells in 50 µL per well and stimulated with 50 µL of SARS-CoV-2 peptide pools (2ug/mL per peptide) in duplicate. R10 with dimethyl sulfoxide (DMSO) (final concentration 0.4%, Sigma) was used as negative control and CEFT ((2 µg/mL, Proimmune)/ Concanavalin A (ConA, 5 µg/mL final concentration, Sigma) were used as positive control antigens. After 16–18 h at 37 °C PBMC were removed and secreted IFN-γ detected using anti-IFN-γ biotinylated mAbs at 1 µg/mL (7-B6-1-biotin, Mabtech) for 2–3 h, followed by streptavidin alkaline phosphatase at 1 µg/mL for 1–2 h (SP-3020, Vector Labs). The plates were developed using BCIP/NBT (5-bromo-4-chloro-3-phosphatase/nitro blue tetrazolium) substrate (Thermo Scientific/Pierce Biotechnology, Rockford, Il) according to the manufacturer's instructions. ELISpot plates were read using an AID ELISpot Reader (v.4.0). Results were reported as spot-forming units (SFU)/$10^6$ PBMC, with a positive control ConA response of >400 SFU/$10^6$ PBMC observed for all assays. Background (mean SFU in negative control wells) was subtracted from antigen stimulated wells to give the final result.

**Reporting summary**. Further information on research design is available in the Nature Research Reporting Summary linked to this article.

## Data availability

The data generated in this study are provided in the Supplementary Information/Source data file. Source data are provided with this paper.

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

## Acknowledgements
Variant B.1.1.7 was isolated and rapidly shared by Kevin Bewley and colleagues within the National Infection Service at Public Health England, Porton Down UK. The customised coronavirus ELISA plates were a gift from Meso Scale Diagnostics, Rockville, MD USA. We thank OUH COVID research nurses and ISARIC. We are grateful for the advice of Professor EC Holmes, University of New South Wales, for advice on the lineage assignment of the isolates used in this study. This work was supported by the UK Department of Health and Social Care as part of the PITCH (Protective Immunity from T cells to Covid-19 in Health workers) Consortium, the UK Coronavirus Immunology Consortium (UK-CIC), National Institute of Health Research (NIHR) and the Huo Family Foundation. It was also directly funded by Department of Health and Social Care (DHSC)/UKRI/NIHR COVID-19 Rapid Response Grant (COV19-RECPLA) and the National Core Study: Immunity (NCSi4P programme) 'Optimal cellular assays for SARS-CoV-2 T cell, B cell and innate immunity'. E.B. and P.K. are NIHR Senior Investigators and P.K. is funded by WT109965MA and NIH (U19 I082360). S.D. is funded by an NIHR Global Research Professorship (NIHR300791). M.C., S.L. and T.T. are funded by a USA FDA grants HHSF223201510104C & 75F40120C00085 Characterization of severe coronavirus infection in humans and model systems for medical countermeasure development and evaluation. A.C.H. and W.J. are supported by University of Oxford Rapid COVID Response Fund, for which the contribution of donors is gratefully acknowledged. D.T.S. and A.J.M are NIHR Academic Clinical Lecturers. J.G.-J. is supported by Ecuadorian National Government Scholarship, M.L.K. is supported by the BBSRC. A.A. is a Wellcome Clinical Research Training Fellow (216417/Z/19/Z). P.K. and M.C. are in the National Institute for Health Research Health Protection Research Unit (NIHR HPRU) in Emerging and Zoonotic Infections (NIHR200907) at University of Liverpool in partnership with Public Health England (PHE), in collaboration with Liverpool School of Tropical Medicine and the University of Oxford. The C-MORE authors' work was supported by NIHR Oxford Biomedical Research Centre, British Heart Foundation (BHF) Oxford Centre of Research Excellence (RE/18/3/34214), United Kingdom Research Innovation. The C-MORE Study is also funded by the Medical Research Council and Department of Health and Social Care/ National Institute for Health Research Grant (MR/V027859/1) ISRCTN number 10980107, as part of the collaborative research programme entitled PHOSP-COVID Post-hospitalisation COVID-19 study: a national consortium to understand and improve long-term health outcomes. The views expressed in this article are those of the authors and not necessarily those of the National Health Service (NHS), the National Institute for Health Research (NIHR), or the Medical Research Council (MRC).

## Author contributions
W.J., M.W.C. and E.B. conceptualised the project. D.T.S., S.D., E.B., P.K., P.G., J.F., C.P.C., K.F., C.D., A.J.P., L.S., A.J.M., S.A.J., A.A., E.A. and H.B. Medawar Laboratory Team, OPTIC Clinical Group, PITCH Study Group and CMORE/PHOSP-C Group established the clinical cohorts and collected and processed the clinical samples and data. W.J. designed and supervised neutralizing antibody work. A.C.H., J.G.-J. and M.L.K performed the neutralizing antibody work, A.R.T. designed and supervised the monoclonal antibody and epitope mapping work, which was carried out and analysed by A.C.H., T.K.T., L.S., K.-Y.A.H., P.R. and M.W.C. designed and supervised the binding antibody work. T.K.T. was carried out by S.L. and T.T. E.B., S.J.D. and P.K. designed and supervised the T Cell work, which was performed by S.A., S.A., A.B., A.S. and T.d.O. provided critical reagents, technical and intellectual expertise. D.T.S., A.C.H., W.J., S.L., T.T., M.W.C., E.B., S.J.D. and S.A. analysed the data. D.T.S., W.J., M.W.C., E.B. and S.J.D. wrote the original draft. D.T.S., A.C.H., W.J., M.W.C., E.B., S.J.D., S.L., T.T. and S.A. reviewed and edited manuscript and figures.

## Competing interests
The authors declare no competing interests.

## Additional information

[1]Peter Medawar Building for Pathogen Research, Nuffield Department of Medicine, University of Oxford, Oxford, UK. [2]Nuffield Department of Clinial Neurosciences, University of Oxford, Oxford, UK. [3]Oxford University Hospitals NHS Foundation Trust, Oxford, UK. [4]James and Lillian Martin Centre, Sir William Dunn School of Pathology, University of Oxford, Oxford, UK. [5]Public Health England, Porton Down, UK. [6]Wellcome Centre for Human Genetics, University of Oxford, Oxford, UK. [7]Peter Medawar Building for Pathogen Research, Department of Paediatrics, University of Oxford, Oxford, UK. [8]Nuffield Department of Medicine, University of Oxford, Oxford, UK. [9]Medical Sciences Division, University of Oxford, Oxford, UK. [10]Translational Gastroenterology Unit, Nuffield Department of Medicine, University of Oxford, Oxford, UK. [11]MRC Human Immunology Unit, MRC Weatherall Institute of Molecular Medicine, University of Oxford, Oxford, UK. [12]Centre for Translational Immunology, Chinese Academy of Medical Sciences, Oxford Institute, University of Oxford, Oxford, UK. [13]Department of Infectious Diseases, Taoyuan General Hospital, Ministry of Health and Welfare, Taoyuan, and Taipei Medical University, Taipei, Taiwan. [14]Oxford Vaccine Group, Department of Paediatrics, University of Oxford, Oxford, UK. [15]NIHR Oxford Biomedical Research Centre, Oxford, UK. [16]Africa Health Research Institute, Durban, South Africa. [17]School of Laboratory Medicine and Medical Sciences, University of KwaZulu-Natal, Durban 4001, South Africa. [18]Max Planck Institute for Infection Biology, Berlin, Germany. [19]KwaZulu-Natal Research Innovation and Sequencing Platform, Durban, South Africa. [20]Centre for the AIDS Programme of Research in South Africa, Durban, South Africa. [21]Department of Global Health, University of Washington, Seattle, WA, USA. [22]Mahidol-Oxford Tropical Medicine Research Unit, Bangkok, Thailand. [23]Centre for Tropical Medicine and Global Health, Nuffield Department of Medicine, University of Oxford, Oxford, UK. [34]These authors contributed equally: Donal T. Skelly, Adam C. Harding. [35]These authors jointly supervised this work: Eleanor Barnes, Miles W. Carroll, William S. James. *Lists of authors and their affiliations appear at the end of the paper. ✉email: william.james@path.ox.ac.uk

## Medawar Laboratory Team

Anthony Brown [1], Senthil Chinnakannan[1], Timothy Donnison[1], Mohammad Ali[1], Patpong Rongkard[1], Matthew Pace[1], Peny Zacharopoulou[1], Nicola Robinson[1], Anna Csala[1], Cathy De Lara[1], Claire L. Hutchings[1], Hema Mehta[1], Lian Ni Lee[1], Matthew Edmans[1], Carl-Philipp Hackstein[1,10], Prabhjeet Phalora[1], Wenqin Li[1], Eloise Phillips[1], Tom Malone[1], Ane Ogbe[1], Cecilia Jay[1] & Timothy Tipoe[1]

## OPTIC (Oxford Protective T cell Immunology for COVID-19) Clinical Group

Lizzie Stafford[8], Thomas Marjot[10], Stavros Dimitriadis[8], Beatrice Simmons[8], Alexandra Deeks[8], Sven Kerneis[3,9], Hibatullah Abuelgasim[3,9], Robert Wilson[3,9], Sarah R. Thomas[3,9], Adam Watson[3,9], Ahmed Alhussni[3,9], Joseph Cutteridge[3,9], Esme Weeks[3,9], Lucy Denly[3,9], Katy Lillie[3,9], Jennifer Holmes[3,9], Philppa C. Matthews[3,8,15] & Denise O'Donnell[8]

## PITCH (Protective Immunity T cells in Health Care Worker) Study Group

Susanna J. Dunachie [1,3,22,23], Lance Turtle[24,25], Thushan de Silva[26,27], Alex Richter[28,29], Christopher J. A. Duncan[30,31], Rebecca P. Payne[31] & Shona C. Moore[24]

[24]HPRU in Emerging and Zoonotic Infections, Institute of Infection, Veterinary and Ecological Sciences, University of Liverpool, Liverpool, UK. [25]Tropical and Infectious Disease Unit, Liverpool University Hospitals NHS Foundation Trust (a member of Liverpool Health Partners), Liverpool, UK. [26]The Florey Institute for Host-Pathogen Interactions and Department of Infection, Immunity and Cardiovascular Disease, University of Sheffield, Sheffield, UK. [27]Sheffield Teaching Hospitals NHS Foundation Trust, Sheffield, UK. [28]Clinical Immunology Service, University of Birmingham College of Medical and Dental Sciences, Birmingham, UK. [29]University Hospitals Birmingham NHS Foundation Trust, Birmingham, UK. [30]Newcastle upon Tyne Hospitals NHS Foundation Trust, Newcastle upon Tyne, UK. [31]Translational and Clinical Research Institute, Newcastle University, Newcastle upon Tyne, UK.

## C-MORE/PHOSP-C Group

Alexander J. Mentzer [6,8], Julian C. Knight[6,15], Mark Philip Cassar[15], Betty Raman[32], Stefan Neubauer[32], Anastasia Fries[8], Nick P. Talbot[33], Nayia Petousi[33], Ling-Pei Ho[11,15], Yanchun Peng[11,12], Tao Dong[11,12], Susana Camara[14,15], Spyridoula Marinou[14,15], Aline Linder[14,15], Syed Adlou[14,15], Mwila Kasanyinga[14,15], Alice Bridges-Webb[14,15], Jennifer Hill[14,15], Laura Silva-Reyes[14,15] & Luke Blackwell[14,15]

[32]Division of Cardiovascular Medicine, Radcliffe Department of Medicine, University of Oxford, Oxford, UK. [33]Department of Physiology, Anatomy and Genetics, University of Oxford, Oxford, UK.

