## [Peer Review File · Nature Communications]

REVIEWER COMMENTS

Reviewer #1 (Remarks to the Author):

The data presented summarizes vaccine responses to SARS-CoV-2 and variants of concern after prime and boost vaccination. Overall, the new material is consistent with what others have found for natural infection or vaccination. The key findings are that the initial response is relatively modest and that viral isolates with mutations in positions 417/484/501 are poorly neutralized by sera from prime vaccination alone. Boost vaccination increased the overall serologic response to SARS-Cov-2 and also enhanced vaccine responses to the variants. The finding that the variant which is spreading most rapidly (B.1.1.7) is not terribly resistant to vaccine responses is reassuring. In addition, they also confirm that T cell responses are less susceptible to the mutations present in the variants than antibodies.

Overall, the manuscript adds valuable information to a growing database on the nature of the human immune response to vaccination against SARS-CoV-2

Reviewer #2 (Remarks to the Author):

In their manuscript Skelly and colleagues characterize antibody responses after natural infection with SARS-CoV-2 and mRNA vaccination in respect to the B.1.1.7 and B.1.351 variants. The manuscript is interesting although not very novel anymore. However, there are several points that need the authors' attention.

Major points

- 1) Many figures are missing positive and negative controls.
- 2) How do the authors know that the difference in neutralization is not caused by differences in viral growth kinetics or minute differences in input virus? Please use an antiviral as control.

Minor points

- 1) Line 51: Please define 'SARS-CoV-2'
- 2) Line 129: 'N501Y', not 'n501Y'
- 3) Line 141: Please provide an appropriate reference. This reference is misplaced.
- 4) Line 150: 'RBD.'
- 5) Line 155: How about Fc-FcR interactions?
- 6) Line 167-172: Some language here is duplicated.
- 7) Line 175: How about Novavax?
- 8) Line 185: Pfizer already showed good efficacy against B.1.351 as well.

- 9) Line 213: Please provide better support for this speculative statement.
- 10) Line 259: Please define 'FFU'.
- 11) Line 275-280: Is there a reason for capitalizing words mid-sentence?
- 12) Line 294: Please define 'MWCO'.
- 13) Line 326: 'SARS-CoV-2', not 'SARS-CoV2'.
- 14) Line 328: Define 'CMV' and 'EBV'.
- 15) Line 407: Define 'IC50'.
- 16) Line 416: 'appear', not 'appears'
- 17) Line 490: 'ELISpot', not 'ELISPOT'
- 18) The language needs to be edited, especially the methods section.

Reviewer #3 (Remarks to the Author):

In their manuscript "Two doses of SARS-CoV-2 vaccination induce more robust immune responses to emerging SARS-CoV-2 variants of concern than does natural infection". Skelly and colleagues compare serum antibody responses and T cell repertoire to SARS-CoV-2 of BioNTech/Pfizer vaccinees with those of asymptomatic or mildly symptomatic COVID-19 convalescents, with emphasis on differential recognition of variants of concern (VOC), the B.1, B.1.1.7, B.1.351 and P.1 variants of SARS-CoV-2B. Take-home message is that the T cell response is pretty diverse and effective also against the VOC, while the antibody-response, at the timepoints measured, is affected significantly, in that antibody titers and efficacy of virus neutralization of VOC, in particular of B1.351, are severely reduced to absent in convalescents and once vaccinated donors, less so in twice vaccinated donors. The manuscript thus provides important information on the protection versus VOC provided by asymptomatic and mild SARS-CoV-2 B infection and single shot BNT162b2 mRNA vaccination, (a) highlighting the added value of vaccination and boost as such, and (b) supporting the need for continued development of adapted vaccines.

In principle, the manuscript is written in a concise and informative way. Nevertheless, at several points information is missing. This might impact on the claims and interpretation of the results, and it precludes publication of the manuscript as it is.

1.) In Fig. 2A and B, it remains a bit unclear, why the difference between antibodies of once and twice vaccinated donors, binding to S and RBD, is "not significant", it might be "less significant"? My recommendation would be to replace the stars by the actual error probabilities ($p < ???$). This adds to a difference of more than 10fold in Fig. 2J, where error probabilities are not indicated at all in the Fig, but only in the text, selectively. Wisely, the authors have separated these data, which they sell as a surrogate to neutralization (line 392), and which shows that both post-prime and post-boost sera are able to block binding of ACE-2 to SARS-CoV-2 S, from the actual neutralization data (Fig. 3D), where they could not detect neutralization of B.1.351 in any of 11 post-prime samples and neutralization of B.1.1.7 only in 4/11 samples (line 446). This discrepancy suggest a dramatic difference in sensitivity between ACE binding inhibition and neutralization assays. On this background, it is not very convincing to argue that ACE2-spike binding inhibition and authentic virus neutralization correlate significantly (line 493ff, Fig. 5A,B). Obviously they do not for those sera which have low antibody titers. Here the claims do not match the data. It might be of relevance, since it remains unclear anyway, how much antibody is required to affect infection in real life.

2.) The inherent question of whether the quantity or the quality of the antibody response changes between the first and second vaccination, and becomes different from the antibody response triggered by natural infection, is not answered by the data provided. It is addressed in a peculiar way, by comparing the ability of six monoclonals binding to each of the 4 major epitopes of SARS-CoV-2 S protein to bind to S of the VOC. The data provided confirm other reports that binding of antibodies to epitopes 1 and 2 of VOC is affected by the mutations, less so binding to epitopes 3 and 4 (Fig. 3A). But the decisive experiment, namely blocking of binding of a monoclonal, recognizing a given epitope, to that epitope, by post-prime, post-boost or convalescent sera has not been done. This would have revealed whether the boost selectively enhances production of antibodies to epitopes 3 and 4, or not, and would have linked the data shown in Fig. 3A with the rest of the paper convincingly.

3.) Last but not least, the authors look at the robustness of the T cell response generated by infection or vaccination, using an interferon ELISPOT assay. Upfront, they find no S protein/peptide-specific memory t cells in unvaccinated and uninfected donors (line 466). This is strange, because such "crossreactive" T cells have been described by others in a significant proportion of the population (e.g. doi: <https://doi.org/10.1101/2021.04.01.21252379>). Is this again a lack of sensitivity or just a reflection of the low number of donors?

Author Responses

Overall, we thank the reviewers for taking the time to provide these constructive and thoughtful comments.

Our point-by-point responses are below. We do hope that our responses meet the requisite standard expected by the reviewers and journal editors and that the paper will be deemed publishable as a result.

REVIEWER COMMENTS

Reviewer #1 (Remarks to the Author):

The data presented summarizes vaccine responses to SARS-CoV-2 and variants of concern after prime and boost vaccination. Overall, the new material is consistent with what others have found for natural infection or vaccination. The key findings are that the initial response is relatively modest and that viral isolates with mutations in positions 417/484/501 are poorly neutralized by sera from prime vaccination alone. Boost vaccination increased the overall serologic response to SARS-Cov-2 and also enhanced vaccine responses to the variants. The finding that the variant which is spreading most rapidly (B.1.1.7) is not terribly resistant to vaccine responses is reassuring. In addition, they also confirm that T cell responses are less susceptible to the mutations present in the variants than antibodies.

Overall, the manuscript adds valuable information to a growing database on the nature of the human immune response to vaccination against SARS-CoV-2

We thank reviewer 1. We agree that the overall messages are relatively reassuring in terms of vaccine immunogenicity for humoral and cellular responses against VOC after two doses.

Reviewer #2 (Remarks to the Author):

In their manuscript Skelly and colleagues characterize antibody responses after natural infection with SARS-CoV-2 and mRNA vaccination in respect to the B.1.1.7 and B.1.351 variants. The manuscript is interesting although not very novel anymore. However, there are several points that need the authors' attention.

Major points

1) Many figures are missing positive and negative controls.

We recognise the requirement for positive and negative controls in experimental work. In general, we believe that we have included requisite controls in the data submitted for this work. We have now improved our description of the controls used in the main body of our manuscript and drawn attention to control data in supplementary information provided. We outline these controls for the referee below.

Negative Controls

Figure 2: A-I: For each of the antigen targets assessed on the MSD multiplex immunoassay platform, we present negative controls, which are pre-pandemic sera collected between 2014 and 2018 (illustrated with black colour). These are described in the methods, results and figure legend. Figure

2J: This assay here was performed as per manufacturer's instructions and internal controls were added to each plate. More extensive SARS-CoV-2 work using this method by our group was recently submitted to the Lancet group and is currently under review ¹.

Figure 3: The supplementary figure 2A shows performance of 4 naïve control donors on the MNA data shown in figure 3. This demonstrates the absence of background neutralization in participants who were not infected or vaccinated. We have updated the manuscript at line 433.

Figure 4: Regarding controls for the T cell data, we have included supplementary Figure 3 which demonstrates the absence of spike-specific responses in vaccine and infection-naïve participants. We have also published details of this assay recently in Nature Communications ². In that paper we utilised multicentre pre-pandemic samples as negative controls to demonstrate the specificity of the assay in three independent laboratories.

Positive controls

The platforms used in this paper– MSD immunoassay, MNA and T cell ELISpot - have been shown to be sensitive for detection of SARS-CoV-2 immune responses in many published papers.

Figure 2: It is clearly accepted in the literature that SARS-CoV-2 infection induces robust IgG induction³. Similarly, BNT162b2 has been shown to induce antibodies against SARS-CoV-2 ⁴ The ability of the MSD platform to reliably detect IgG and demonstrate ACE2 binding as a proxy for neutralization has been previously published ⁵ and the assays were carried out as per manufacturer's instructions. Consequently, we believe the inclusion of SARS-CoV-2 positive samples on the MSD platform provides an adequate positive control. In addition, all plates have an internal calibration curve. For the ACE2 inhibition binding assay the NISBC international standard was added to each plate ¹.

Figure 3: The B panel shows the performance of NIBSC 20/130 plasma on our MNA ⁶. The purpose of this international standard is for comparison across labs and to ensure calibration. The designed homotypic NT50 of this is around 1/1000, consistent with our results.

Figure 4: Every ELISpot plate included ConA as a positive control, which is shown in Figure 4D. We have added to the methods section "with a positive control ConA response of >400 SFU/106 PBMC observed for all assays". The performance of this assay with SARS-CoV-2 positive serum samples in our hands has been previously published in Nature Communications (Ogbe et al)⁷. We have now entered a reference for this paper at line 355 of methods.

We do hope these responses and editing of the text to match it are sufficient to allay the reviewer's legitimate concerns.

2) How to the authors know that the difference in neutralization is not caused by differences in viral growth kinetics or minute differences in input virus? Please use an antiviral as control.

1 https://papers.ssrn.com/sol3/papers.cfm?abstract_id=3812375

2 <https://www.nature.com/articles/s41467-021-21856-3>

3 [https://www.thelancet.com/journals/laninf/article/PIIS1473-3099\(20\)30634-4/fulltext](https://www.thelancet.com/journals/laninf/article/PIIS1473-3099(20)30634-4/fulltext)

4 <https://www.nejm.org/doi/10.1056/NEJMoa2027906>

5 <https://doi.org/10.1016/j.jcv.2020.104572>

6 https://nibsc.org/products/brm_product_catalogue/detail_page.aspx?catid=20/130

7 <https://doi.org/10.1038/s41467-021-21856-3>

The assay assesses neutralization by the counting of discrete foci, with each focus representing an individual infectious event (See image below). Input virus is standardised for all variants to ensure a consistent number of infectious virus particles are present in each well, regardless of variant. To control for inevitable small fluctuations in the biological assay, all foci counts are normalized to internal serum-free controls present in each plate, where this “virus-only” condition is taken to be 100% of the expected foci in the absence of neutralization.

Because the assay uses a viscous overlay to prevent progeny virus spreading beyond the bounds of a focus, differences in growth kinetics will only alter the size of a focus and will not lead to an increase in total number of foci. The differences in growth kinetics of the variants have been accounted for by altering the post-infection incubation time, to produce foci that are of equivalent size in all conditions.

The combination of our NIBSC reference plasma, and differential responses to monoclonal antibodies targeting distinct epitopes, demonstrate that neutralization disparity between variants is not explained by differences in growth kinetics or input virus as the causative variable. Neutralization does not drop or increase uniformly with specific variants, but rather is highly dependent on the epitope(s) of the neutralizing sample. The apparent uniformity of potency reduction in sera tested against variants is indicative of an innately polyclonal response, akin to averaging the responses seen with individual antibodies.

We believe the combination of naïve (uninfected, unvaccinated) sera, and an internationally recognised reference plasma (NIBSC 20/130), act as robust negative and positive controls, respectively. Furthermore, our assay for detecting neutralisation is specifically designed to assess neutralizing antibodies and is not suitable as a readout for antivirals. This is due to the mechanism by which they act. In general, neutralization occurs prior to the virion entering the cell and is assessed as an all-or-none event (either a focus is produced, or it is not). In contrast, most antivirals don't act until after a virion has entered the cell and are typically assessed by a reduction in output virus. However, in a focus-forming assay, even a 50% reduction in progeny virus would still result in the production of a focus, and thus makes the comparison to sera neutralization inappropriate.

MNA plate images for B, B.1.1.7 and B.1.351

Minor points

1) Line 51: Please define 'SARS-CoV-2'

Fixed

2) Line 129: 'N501Y', not 'n501Y'

Fixed

3) Line 141: Please provide an appropriate reference. This reference is misplaced.

Fixed

4) Line 150: 'RBD.'

Fixed

5) Line 155: How about Fc-FcR interactions?

We agree that these interactions are potentially important in antibody-mediated protection. We now include the following addition in the text to recognise this at Line 159 - 'Antibodies may also offer protection via fragment crystallizable (FC)-FC receptor interactions and harnessing of innate immune function. Diverse antibody dependent macrophage, neutrophil, complement and natural killer cell functions have been demonstrated after SARS-CoV-2 infection and vaccination.'

6) Line 167-172: Some language here is duplicated.

This text has been improved

7) Line 175: How about Novavax?

We had not included Novavax in this sentence as its phase III interim results had not been published (and remain unpublished to the best of our knowledge). Moreover, in its reported interim results it doesn't quite reach 90% efficacy. We do, however, acknowledge its reported success against the B.1.351 variant in these interim results on line 186.

8) Line 185: Pfizer already showed good efficacy against B.1.351 as well.

These data have only emerged between submission of this manuscript and the referee's comments. Although these data remain unpublished other than the Pfizer statement, we have included reference to it now as it reports 100% efficacy in contrast to the other vaccines.

9) Line 213: Please provide better support for this speculative statement.

This statement has now been modified and a supporting reference entered.

10) Line 259: Please define 'FFU'.

Done

11) Line 275-280: Is there a reason for capitalizing words mid-sentence?

Now addressed.

12) Line 294: Please define 'MWCO'.

Fixed

13) Line 326: 'SARS-CoV-2', not 'SARS-CoV2'.

Fixed

14) Line 328: Define 'CMV' and 'EBV'.

Now Done

15) Line 407: Define 'IC50'.

Done

16) Line 416: 'appear', not 'appears'

Fixed

17) Line 490: 'ELISpot', not 'ELISPOT'

Fixed

18) The language needs to be edited, especially the methods section.

Done

Reviewer #3 (Remarks to the Author):

In their manuscript "Two doses of SARS-CoV-2 vaccination induce more robust immune responses to emerging SARS-CoV-2 variants of concern than does natural infection". Skelly and colleagues compare serum antibody responses and T cell repertoire to SARS-CoV-2 of BioNTech/Pfizer vaccinees with those of asymptomatic or mildly symptomatic COVID-19 convalescents, with emphasis on differential recognition of variants of concern (VOC), the B.1, B.1.1.7, B.1.351 and P.1 variants of SARS-CoV-2B. Take-home message is that the T cell response is pretty diverse and effective also against the VOC, while the antibody-response, at the timepoints measured, is affected significantly, in that antibody titers and efficacy of virus neutralization of VOC, in particular of B1.351, are severely reduced to absent in convalescents and once vaccinated donors, less so in twice vaccinated donors. The manuscript thus provides important information on the protection versus VOC provided by asymptomatic and mild

SARS-CoV-2 B infection and single shot BNT162b2 mRNA vaccination, (a) highlighting the added value of vaccination and boost as such, and (b) supporting the need for continued development of adapted vaccines.

In principle, the manuscript is written in a concise and informative way. Nevertheless, at several points information is missing. This might impact on the claims and interpretation of the results, and it precludes publication of the manuscript as it is.

1.) In Fig. 2A and B, it remains a bit unclear, why the difference between antibodies of once and twice vaccinated donors, binding to S and RBD, is "not significant", it might be "less significant"? My recommendation would be to replace the stars by the actual error probabilities ($p < ???$). This adds to a difference of more than 10fold in Fig. 2J, where error probabilities are not indicated at all in the Fig, but only in the text, selectively. Wisely, the authors have separated these data, which they sell as a surrogate to neutralization (line 392), and which shows that both post-prime and post-boost sera are able to block binding of ACE-2 to SARS-CoV-2 S, from the actual neutralization data (Fig. 3D), where they could not detect neutralization of B.1.351 in any of 11 post-prime samples and neutralization of B.1.1.7 only in 4/11 samples (line 446). This discrepancy suggest a dramatic difference in sensitivity between ACE binding inhibition and neutralization assays. On this background, it is not very convincing to argue that ACE2-spike binding inhibition and authentic virus neutralization correlate significantly (line 493ff, Fig. 5A,B). Obviously, they do not for those sera which have low antibody titers. Here the claims do not match the data. It might be of relevance, since it remains unclear anyway, how much antibody is required to affect infection in real life.

With regards figure 2A-I, we understand the reviewer's desire to see *P* values on the figure and have adjusted the figure accordingly. All groups were compared but we have chosen to only indicate significant post-hoc comparisons, rather than also reporting non-significant post-hoc comparisons due to the resultant figure being somewhat cluttered. We state in the legend that only significant *P* values are displayed.

Based on our previous formatting of the panels, the lack of a statistically significant difference between post-prime and post-boost samples in 2A (spike) and 2B (RBD) was indeed somewhat surprising. We have used a y axis with no break for both of these graphs for this resubmission. We think it is now clearer that there is variability in the post-prime response, which inevitably affects the test for significance. We apologise to the reviewer if the presentation had previously made interpretation difficult. The combination of a stringent Kruskal Wallis test with relatively low sample numbers (particularly the post-boost group with n=11) contributes to the absence of significance between the groups. We note for the reviewer that for spike and RBD binding results, the post hoc P value of comparisons between the post prime and post boost sample is >0.999999 . Thus, they are not 'less significant' by this statistical approach. However, when the data are log-transformed, multiple comparisons tests after one-way ANOVA are highly significant for both spike and RBD. For consistency, we have chosen to stick with the original way the data was presented.

We are in agreement with the referee that we had not previously reported or displayed the multiple comparisons for 2J (ACE2 binding inhibition) in a suitably comprehensive way. All significant statistical differences within the pre and post samples are now clearly indicated on the figure with description in the figure legend. Similarly, significant Mann-Whitney tests between the post-prime and post-boost groups for each spike variant are indicated on the figure and described in the figure legend.

Regarding ACE2 blocking/MNA post-prime differences, we believe that the apparent discrepancy between neutralization data and the corresponding ACE2 inhibition is in line with expectations considering the mechanism of the two assays. The ACE2 inhibition assay is, at its lower limits, detecting blocking of a small proportion of available spikes. However, it remains an open question as to what proportion of SARS-CoV-2 S trimers must be blocked in order to achieve neutralization of a virion. Naturally, any minimum threshold beyond a handful of trimers would result in a disparity between the assays – akin to that shown in our own data. It seems likely that what appears to be increased sensitivity in the ACE2 blocking assay, is in fact detection of the sub-neutralizing range of spike binding.

Whilst we recognise that neutralization assays will fail to detect neutralizing antibody activity in samples that contain sub-neutralizing concentrations of neutralizing antibodies by definition, this would not detract from the primary observation of broad NT50 differences between cohorts. Furthermore, when conducting statistical analyses, we were diligent in setting NT50s to the assay's limit of detection for all samples that failed to produce detectable neutralization. That said, we are reassured that the sensitivity of our MNA is well within the range of comparable assays in the literature, as confirmed by the NIBSC reference plasma (Figure 3B).

2.) The Inherent question of whether the quantity or the quality of the antibody response changes between the first and second vaccination, and becomes different from the antibody response triggered by natural infection, is not answered by the data provided. It is addressed in a peculiar way, by comparing the ability of six monoclonals binding to each of the 4 major epitopes of SARS-CoV-2 S protein to bind to S of the VOC. The data provided confirm other reports that binding of antibodies to epitopes 1 and 2 of VOC is affected by the mutations, less so binding to epitopes 3 and 4 (Fig. 3A). But the decisive experiment, namely blocking of binding of a monoclonal, recognizing a given epitope, to that epitope, by post-prime, post-boost or convalescent sera has not been done. This would have revealed whether the boost selectively enhances production of antibodies to epitopes 3 and 4, or not, and would have linked the data shown in Fig. 3A with the rest of the paper convincingly.

We thank the reviewer for this incisive comment. As they rightly imply, the aim of our paper was not to undertake a detailed qualitative analysis of the distribution of epitope specificities and affinities in the sera of donors, after recovery or vaccination, and our results do not answer such interesting questions. Indeed, such an analysis would be extremely complex and lies outside the scope of this paper. The mAbs were simply used here to demonstrate that (as expected) antibodies to the conserved surface of the RBD still neutralised, those to the variable bit did not, and we apologize if this was unclear. We have tried to clarify this in our revision. Our experiment simply shows that there is a real difference between the groups studied, and points to the neutralization reactivities being related to protection. This remains a very relevant observation, given the SA experience, in which natural infection did not protect against symptomatic infection with variant, and the more recent incident in a Kentucky nursing home, reported by CDC.

The specific experiment suggested by the reviewer would in principle be interesting, but not be quite as decisive as implied (more detailed argument is provided at *, below). In any case, the issue is moot, as there remains insufficient serum from most of the donors to enable this experiment to be undertaken. We respectfully request to be excused from attempting this additional work, therefore, on the grounds of both relevance and practicability.

* We have undertaken extensive monoclonal antibody blocking experiments for our earlier work on the response to vaccination in Tan et al ⁸. A change in the ability to block monoclonal antibody binding can be either quantitative (a change in the affinity or quantity of particular antibodies) or qualitative (a change in the pattern of antibodies), or any combination of the two. Blocking experiments can be very complex to interpret and it is common to see "one-way" blocking where a given antibody A blocks B, but B does not block A - for reasons that are not necessarily clear. We think the only way to answer the qualitative side of this would be to analyse the pattern of reactivities by Cryo EM of Fabs isolated from sera. This would involve a series of major experiments, which would then be very difficult to interpret given the variation between individuals (as recently shown by Jesse Bloom's Lab ⁹).

3.) Last but not least, the authors look at the robustness of the T cell response generated by infection or vaccination, using an interferon ELISPOT assay. Upfront, they find no S protein/peptide-specific memory t cells in unvaccinated and uninfected donors (line 466). This is strange, because such "crossreactive" T cells have been described by others in a significant proportion of the population (e.g.doi: <https://doi.org/10.1101/2021.04.01.21252379>). Is this again a lack of sensitivity or just a reflection of the low number of donors?

It is correct that several other leading laboratories are using ELISpot assays that detect cross-reactive T cell responses to S1 and S2 in a proportion of unvaccinated and uninfected donors. Our previous published work, Ogbe et al 2021,¹⁰ demonstrated how we have developed a range of T cell assays with varied sensitivity and specificity to detect such cross-reactive T cells to S1 and S2. We have selected a highly specific ELISpot assay using cryopreserved cells and a combination of peptide concentration and incubation times that result in negligible cross-reactivity in unexposed populations to allow more accurate monitoring of response to natural infection and vaccination, in contrast to our 7-day CTV proliferation assay which detects such cross-reactivity to S1 and S2 in the majority of unexposed populations (Ogbe et al 2021). We have now harmonised this ELISpot assay

⁸ <https://www.nature.com/articles/s41467-020-20654-7>

⁹ <https://science.sciencemag.org/content/371/6531/850.abstract>

¹⁰ <https://doi.org/10.1038/s41467-021-21856-3>

across 7 UK laboratories with similar negligible responses to S1 and S2 in unexposed people (SOP available in supplement of Angyal et al¹¹), and we consider this an advantage for focussing on vaccine responses. The study link provided by the referee utilises flow cytometry approaches to identify these cells, consistent with our results. We have added text to the manuscript results section to better explain this “Spike-specific T cells could not be detected in unvaccinated SARS-CoV-2 unexposed HCW (Figure S3), in keeping with our previously published work demonstrating use of a highly specific IFN- γ ELISpot assay with the selected peptide concentration and incubation time resulting in negligible responses detected to spike peptides in unexposed subjects”.

¹¹ <http://dx.doi.org/10.2139/ssrn.3812375>

REVIEWER COMMENTS

Reviewer #2 (Remarks to the Author):

The authors have addressed my comments.

Reviewer #3 (Remarks to the Author):

It is an interesting rebuttal, defining the limitations of the presented work more clearly and discussing it much better in the frame of the current landscape of literature on adaptive SARS-CoV-2 immune responses. From my point of view, the presentation of data and the statistical evaluations are now much more comprehensive and the somewhat disparate "chapters" of the manuscript much better connected. The basic claims have been adapted and are now clearly supported by the data.